# A Surface-Enhanced Raman Spectroscopy-Based Biosensor for the Detection of Biological Macromolecules: The Case of the Lipopolysaccharide Endotoxin Molecules

**DOI:** 10.3390/ijms241512099

**Published:** 2023-07-28

**Authors:** Giulia Rusciano, Angela Capaccio, Antonio Sasso, Alessandro Capo, Carlos Murillo Almuzara, Maria Staiano, Sabato D’Auria, Antonio Varriale

**Affiliations:** 1Department of Physics “E. Pancini”, University of Naples “Federico II”, 80126 Naples, Italy; giulia.rusciano@unina.it (G.R.); angela.capaccio@isa.cnr.it (A.C.); antonio.sasso@unina.it (A.S.); 2Institute of Food Sciences (ISA), CNR, 83100 Avellino, Italy; alessandro.capo@isa.cnr.it (A.C.); carlosmualsola@gmail.com (C.M.A.); maria.staiano@isa.cnr.it (M.S.); 3Department of Biology, Agriculture and Food Sciences, CNR, 00185 Rome, Italy; 4Institute of Food Sciences, URT-CNR at Department of Biology, University of Naples “Federico II”, 80126 Naples, Italy; antonio.varriale@isa.cnr.it

**Keywords:** LPS, SERS, Raman spectroscopy, antibody, biosensor, Nano immunoassay

## Abstract

The development of sensitive methods for the detection of endotoxin molecules, such as lipopolysaccharides (LPS), is essential for food safety and health control. Conventional analytical methods used for LPS detection are based on the pyrogen test, plating and culture-based methods, and the limulus amoebocyte lysate method (LAL). Alternatively, the development of reliable biosensors for LPS detection would be highly desirable to solve some critical issues, such as high cost and a long turnaround time. In this work, we present a label-free Surface-Enhanced Raman Spectroscopy (SERS)-based method for LPS detection in its free form. The proposed method combines the benefits of plasmonic enhancement with the selectivity provided by a specific anti-lipid A antibody (Ab). A high-enhancing nanostructured silver substrate was coated with Ab. The presence of LPS was quantitatively monitored by analyzing the changes in the Ab spectra obtained in the absence and presence of LPS. A limit of detection (LOD) and quantification (LOQ) of 12 ng/mL and 41 ng/mL were estimated, respectively. Importantly, the proposed technology could be easily expanded for the determination of other biological macromolecules.

## 1. Introduction

Lipopolysaccharides (LPS) are endotoxins, harmful and toxic inflammatory stimulators, present in the membrane of Gram-negative bacteria. From a structural point of view, LPS is a well-known molecule. It is organized into three structural domains: the core oligosaccharide, the O antigen, and the lipid A, which is the last and most preserved one among bacteria and responsible for biological activity and toxicity. The death and/or disintegration of this class of bacterial cells cause the release of active cell components, leading to organ failure, septic shock, and even death. In fact, upon interaction with human cell membrane receptors (toll-like receptors, such as TRL4), LPS activates the production of inflammatory cytokines and chemokines, which contribute to the inflammatory process with the severe effects of septic shock syndrome [1,2]. In addition, the ubiquitous distribution of bacteria in the environment causes a constant detachment of free LPS molecules from the cell walls, which also needs to be detected. Consequently, it is crucial to have real-time monitoring of the presence of free LPS molecules in environmental, food, drug, and clinical treatment settings.

The most conventional methods used for the detection of LPS—approved by the FDA and the European Medicines Agency (EMA)—are based on the rabbit pyrogen test (RPT) and the limulus amoebocyte lysate (LAL) [3,4,5]. Although these methodologies are recognized as valid, they have some disadvantages owing to their costs, being time-consuming, and the involvement of animals.

Compared to the RPT technique, LAL tests are more quantitative and sensitive (limit of detection (LOD)~10^−9^ g/mL) [6]. However, it is important to highlight that the sensitivity of the assay is strongly affected by the experimental conditions, such as temperature, pH values, and reaction time. In addition, different configurations of ELISA tests have been developed and used in clinical practice for the detection of LPS [7]. The ELISA test has the advantage of being highly sensitive, but it is time-consuming, thus not allowing critical decisions to be made. Consequently, there is an urgent need to develop a faster method for detecting the presence of LPS. In this context, among the different approaches [8,9,10], surface enhanced Raman spectroscopy (SERS) benefits from the strong plasmonic enhancement of the Raman signal, together with the “fingerprint” character of Raman spectra [11,12,13]. Even more important, SERS signals avoid the typical drawbacks of fluorescence-based approaches, mainly related to photobleaching and photoblinking; in such a way, prolonged and repeated measurements become possible, with consequent enhancement of the obtainable signal-to-noise ratio. So far, the SERS technique has been used for the study of a wide range of biomolecules [14,15,16,17], including free LPS molecules from E. coli and K. pneumoniae present in aqueous solution [18]. In particular, a completely label-free protocol was chosen, with LPS allowing it to be adsorbed on the surface of 50 nm AuNPs. Accordingly, LPS detection was performed taking advantage of the direct observation of the intrinsic LPS spectral features. However, LPS detection in complex environments requires the development of more sophisticated approaches to assure both proper sensitivity and selectivity.

In common SERS-based immunoassay protocols, a “sandwich” scheme based on selective antibody-antigen binding is used [19]. In such a case, captured primary antibodies are bound to a gold surface (even a flat surface) to produce an immune substrate. SERS probes are commonly constituted of gold nanoparticles (NPs) bound to secondary antibodies and labeled with a selected Raman reporter, typically constituted by molecular species exhibiting a high Raman cross-section and a well-known Raman spectrum. When target antigens are added to the immune substrate, they are captured by the antibodies present on it and subsequently recognized by the SERS probes. Quantitative measurements of antigen concentrations are finally performed on the basis of the SERS spectrum corresponding to the Raman reporter. Such an approach has been successfully used for the sensitive and selective detection of numerous biomolecules. Nevertheless, the complexity of the sandwich-based modality usually involves long reaction times, and it can suffer from non-specific adsorption on the Raman reporter. More recently, alternative SERS-immunoassay protocols have been developed based on the detection of the shift in selected bands of the Raman reporter (“frequency-shift SERS sensing”) [20]. Quantitative information on the analyte concentration is obtained by measuring the frequency shifts in intense SERS peaks of a Raman reporter upon binding events with the analyte, driven by mechanical transduction [21], charge transfer [22], or the simple modification of the Raman reporter environment [23]. Intriguingly, perturbations to the reporter SERS spectra can involve not only the peak shifts but also the change of the peak shape, as demonstrated in ref. [24]. Notably, binding events can be efficiently detected by using relatively large molecules as Raman reporters, as demonstrated in the case of SARS-CoV [25].

In this work, we present a frequency-shift SERS-based biosensor for the detection of LPS by using the anti-lipid A polyclonal antibody (Ab) as a Raman reporter. In particular, a high-enhancing silver substrate was properly coated by such commercial polyclonal antibodies, which were able to bind free LPS molecules. Therefore, a strong SERS signal ascribable to Ab was detected. Finally, the presence of LPS was quantitatively monitored by following the changes in the Ab SERS spectra induced upon LPS binding. Such an antibody-mediated label-free detection scheme, previously demonstrated for other analytes [21,26,27], avoids the use of the secondary antibody required in the conventional ELISA test and the development of an ad hoc Raman reporter, therefore greatly simplifying and speeding up the detection protocol. The obtained data show the possibility of detecting the presence of LPS in solution with a limit of detection (LOD) of 12 ng/mL and a limit of quantification (LOQ) of 41 ng/mL. Importantly, the proposed technology can be easily expanded to the determination of other biological macromolecules.

## 2. Results and Discussion

The first step was to identify a suitable molecular recognition element (MRE) able to specifically bind to LPS. The lipid A polyclonal antibody able to selectively bind LPS from E. coli O157 was identified.

### 2.1. LPS Detection in Aqueous Solution

A SERS substrate exhibiting nine wells was prepared ad hoc as reported in the Material and Methods section (see step 1 described in paragraph 3.3). Each well was filled with 20 μL of LPS solutions at concentrations ranging from 0.25 µg/mL to 10 µg/m and incubated for two hours, allowing for the absorption of the substrate, before the removal of the not adsorbed molecules by an absorbing paper. The prepared substrate was therefore analyzed by SERS. It is worth noting that in the studied concentration range, the spontaneous Raman spectrum provided non-appreciable signals At a fixed concentration, 5 raster scans were performed centered at randomly selected points of the well under investigation. At each scan, N = 100 spectra were recorded in a 20 × 20 µm area. Figure 1a reports the averaged SERS spectra obtained at the concentration values ranging from 0.25 to 10 µg/mL, as obtained by considering all the 500 spectra acquired in the considered well.

Clearly, spectra averaging over a total of 5 × N = 500 spectra helped us to take into account the spatial heterogeneity due to both hot spots and molecule distribution on the substrate. Figure 1b shows the average SERS signal intensity of the peak at 2900 cm^−1^ at the concentration of [LPS] = 1.25 µg/mL obtained in the 5 maps. The relative variability of the SERS signal on a single scan is around 19%. Notably, the average SERS intensities of each scan are consistent, and by averaging them, the SERS signal fluctuation can be reduced to about 12%.

The assignment of the observed bands is well reported in the literature [28,29,30]. To investigate the sensitivity of this approach, for each analyzed concentration, we calculated the signal-to-noise ratio (SNR) and the relative standard deviation of the peak corresponding to the more isolated band at 2900 cm^−1^ and fitted the low concentration part with a linear fit (see Figure 1c,d). From this procedure, we estimated a C_min_ of about 3 ng/mL, which is close to that presented in similar studies [18].

### 2.2. Analysis of Ab@c-SERS Substrate

As previously anticipated, the detection of LPS in complex matrices, i.e., in real samples, requires both high sensitivity and selectivity. The achievement of these goals requires the development of proper detection strategies, taking advantage of both the plasmonic amplification of signals and the unique selectivity provided by the antigen-antibody binding step. For this reason, we developed a label-free SERS-based immunoassay approach for the detection of LPS. The basic idea of this strategy is to first detect the SERS spectrum of Ab alone and, then, study the variations of this spectrum once LPS binds to Ab. Thus, these modifications are quantitatively correlated with the LPS concentration.

In order to optimize the LPS detection approach, we accurately characterized the SERS substrate coating with Ab. As previously mentioned, to prevent unspecific binding events of spurious molecules to the SERS substrate, the Ab-coated silver substrate was incubated with a blocking solution (BS), as described. Reasonably, due to the high BS affinity with Ag, most of the hot spots not previously occupied by Ab were filled with BS.

Figure 2 reports the average SERS spectrum corresponding to the prepared Ab@c-SERS substrate. The average was performed by acquiring 100 spectra by raster scanning in a 20 × 20 µm region. Notably, such average signals are quite reproducible, which constitutes the fundamental prerequisite for the effectiveness of our protocol. The observed SERS signals are dominated by the parts that are placed very close to the metallic surface. Moreover, the Raman modes perpendicular to the surface are preferably enhanced. Clearly, the presence of relatively broad bands in the spectrum of Figure 2, rather than sharp peaks, reflects the complexity of the sample and, to a certain extent, it makes a precise assignment of the observed bands difficult. However, some tentative assignments can be reasonably performed. For instance, the band around 640 cm^−1^ is due to C-C-C out of plane bending, while the band around 840 cm^−1^ may be due to the same vibrational mode of the C-H bond. The bands around 1250 and 1659 cm^−1^ can be due to C-N and C=O stretching, respectively, while the sharp peak at 1001 cm^−1^ is due to phenylalanine. Finally, the bands around 1449 and 3058 cm^−1^ can both be assigned to the C-H bond. In particular, the band around ~1449 cm^−1^ can be assigned to CH_3_ asymmetrical bending, while the high wavenumber bands can be assigned to the overlapping contributions of CH_2_ (at 2883 cm^−1^) and CH_3_ (at 2933 cm^−1^) symmetric stretching. It is worth noticing that the two latter bands dominate the Ab@c-SERS spectrum. In this regard, two important issues can be highlighted: (a) the c-SERS substrate has a quite wide spectral response; (b) it assures a significant signal enhancement also for bands at relatively high wavenumbers.

In order to optimize the LPS detection capability, particular care was taken to find the optimal coating of the SERS substrate in terms of Ab and BS. Although the use of a BS is unavoidable to prevent unspecific binding events of spurious molecules, BS binding to the Ag substrate can give rise to interfering SERS peaks that overlap with the Ab spectrum. Ab@c-SERS surface coverage optimization was carried out following the procedure illustrated in step 1–2–3 described in paragraph 3.3 and incubating some of the wells with three different Ab concentrations in the presence of BS solution, which were 0.1 μg/mL, 1 μg/mL, and 10 μg/mL, keeping fixed the BS concentration (sample “Ab+BS”). As reference, three different wells coated with Ab solution, at the same concentration without BS solution (sample “Ab”) were prepared. Finally, a last well was incubated with BS (sample “BS”). For each well, SERS analysis was performed by acquiring 400 spectra via raster scans in 20 × 20 µm^2^ regions and processing the acquired spectra by PCA.

Figure 3 reports the results of the analysis. For each Ab solution concentration, the PC1-PC3 score plot is shown (no significant data differentiation can be appreciated along the PC2 component). As a common feature, score points corresponding to spectra of pure Ab coating (blue dots) for all the three concentration values here investigated appear well separated from points corresponding to BS spectra (red dots). More intriguing is, instead, the case of the co-presence of the two molecular species (“Ab+BS”, yellow dots). In fact, at the lowest concentration value (0.1 µg/mL), yellow dots tend to be spread in the PC1-PC3 plane, partially overlapping the regions occupied by red and blue dots. This result confirms the fact that both molecules (Ab and BS) have comparable probabilities of occupying the hot spots of the SERS substrates. However, by increasing the Ab concentration values, the hot spots free to host BS reduce, and “Ab” and “Ab+BS” samples tend to cluster in the same region of the score plot. This is particularly true for the highest Ab concentration value (10 µg/mL), where yellow and blue points are almost completely overlapped. We speculate that this outcome can be read as a saturation effect of the hot spots by Ab molecules, with only a small residual fraction occupied by BS molecules. From a spectral point of view, this is mirrored by the indistinguishability of the “Ab” spectrum (indicated as control) and the “Ab+BS” spectra, as highlighted in Figure 3c.

### 2.3. LPS Detection by Ab@c-SERS

Once the optimized Ab coverage was established (10 µg/mL), Ab@c-SERS substrates were incubated in the presence of LPS. Therefore, we proceeded with the selective detection of LPS and the evaluation of its limit of detection (LOD) and limit of quantification (LOQ). For this purpose, we acquired SERS spectra on Ab@c-SERS substrate, varying LPS concentration values in the range from 1 ng/mL to 10 µg/mL. Such measurements were performed by following the procedure described in the Materials and Methods section (see step 1–2–3 described in paragraph 3.3). In Figure 4a, the averaged SERS spectra at the selected concentration values are shown, including the control sample (Ab@c-SERS substrate with [LPS] = 0 µg/mL).

A rapid inspection of these spectra reveals a significant difference with respect to the case of Ab@c-SERS substrates, denoting an evident effect of the interaction with Ab-LPS. In particular, two interesting trends can be observed. First, the broad band at ~1660 cm^−1^ appears split into two bands, with the rise of a sharp band around 1600 cm^−1^. Next, it is possible to appreciate a blue shift of the bands around 2900 cm^−1^, with the spectral envelope maximum passing from 2933 cm^−1^ for [LPS] = 0 (Ab@c-SERS spectrum) to 2922 cm^−1^ at [LPS] = 10 µg/mL (Figure 4b). The connection of these perturbing effects with LPS concentration values has been investigated by following two different approaches, as described below.

#### 2.3.1. Analysis in the Low Wavenumber Region

As a first step, we analyzed the 1100–1800 cm^−1^ spectral region. To explain the modification of the observed spectra, it is reasonable to exclude a direct contribution of LPS binding to Ag nanostructure in view of the saturation of hot spot occupation by Ab molecules discussed in the previous paragraph. The spectral region considered herein exhibits numerous bands, mainly related to C-C and C-H bonds. Each of these bands could, in principle, be affected by LPS binding to Ab in a way which is difficult to predict a priori, considering the complexity of the two molecular structures. Moreover, due to the band overlapping of Ab and LPS, the analysis of this region in terms of data fitting by a multi-peak function results in strong model dependence. Therefore, we considered a multivariate statistical analysis of the spectra by PCA more appropriate. This analysis was based on 400 spectra acquired at different sites on the SERS surface for each LPS concentration value. In Figure 5a, we report the score plot projection in the PC1-PC3 plane. As it can be seen, dots corresponding to a given LPS concentration value are spread in a relatively wide cloud in this plane, suggesting a relatively high variability of the spectra. Nevertheless, it is possible to appreciate a global shift of clouds according to the direction indicated by the yellow arrow passing from the lowest to the highest concentration values. This behavior can be better recognized in Figure 5b, which reports the projections along the PC3 axes of the centroids of each cloud as a function of LPS concentration values. In addition, as expected, the variability evidenced by PCA can be mainly attributed to the band at ~1600 cm^−1^, which constitutes the prominent spectral feature of the PC3 vector (Figure 5c). Finally, the PC1-PC3 loading plot (Figure 5d) reveals the strong correlation of the variability of this peak with the shift of the score’s clouds.

#### 2.3.2. Analysis in the High Wavenumber Region

As a final step, we analyzed the variations of Ab@c-SERS spectra induced by LPS binding to Ab in the region between 2650 cm^−1^ and 3150 cm^−1^. In this spectral range, a frequency-shift SERS approach was used. From a spectral point of view, this region is dominated by features due to both symmetric and asymmetric stretching from the CH_2_ (left wing) and CH_3_ (right wing) groups. Notably, we verified that the resulting envelope can be effectively described by the overlapping of four Gaussian peaks. This point is illustrated in Figure 6a, which reports the deconvolution of the average spectrum obtained at [LPS] = 10 µg/mL. In particular, the blue dashed lines correspond to the single Gaussian peaks, while the continuous red line corresponds to the fitted spectral envelope. A similar procedure was applied to all the LPS investigated concentration values. Intriguingly, two of the fitting output parameters exhibit a clear correlation with LPS concentration values. This is the case of the centers c_1_ and c_2_ of two Gaussian peaks lying in the left wing of the total envelope. These bands can be assigned to CH_2_ and CH_3_ stretching bands, centered around 2880 cm^−1^ and 2933 cm^−1^, respectively. Figure 6b reports the shifts Δ_2933_ of the most prominent peak of center c_2_ with respect to the naked Ab@c-SERS substrate case (i.e., [LPS] = 0 ng/mL) as a function of LPS concentration values. Interestingly, again, a quite linear behavior for [LPS] < 1 µg/mL is observed, after which saturation is achieved. On the contrary, peaks associated with CH_2_ stretching do not exhibit a clear correlation with LPS concentration values, although a blue shift is observed between [LPS] = 0 ng/mL and [LPS] = 10^4^ ng/mL. Clearly, a numerical simulation could help shed light on this experimental outcome. However, it is outside the scope of the present investigation. The experimental trends observed for shift Δ_2933_ can be used to get quantitative information to estimate the LOD and LOQ. For this purpose, the SNR was calculated for the low concentration data according to the relation:SNR = ∆_2933_/σ_Δ_, (1)
being σ_Δ_ the error on the estimated shift value, resulting from the fitting of the spectral features. Finally, a linear fitting of data was performed, from which the LOD was evaluated by extrapolating the LPS concentration at which SNR = 3. Similarly, LOQ was evaluated by considering a SNR of 10:1. Notably, the estimated LOD and LOQ were 12 ng/mL and 41 ng/mL, respectively.

## 3. Materials and Methods

### 3.1. Materials

Commercial lipid A LPS polyclonal antibodies were purchased from Invitrogen (Invitrogen, Waltham, MA, USA, PA1-73178). Bovine serum albumin (BSA) (810535) and LPS were purchased from Sigma-Aldrich, St. Louis, MI, USA (L6529).

### 3.2. Preparation of Coral-like SERS Substrates

SERS substrates were prepared according to the procedure described in refs. [31,32]. Briefly, an adhesion bilayer composed of a 10 nm thick Au layer over-imposed on a 3 nm thick Cr layer was first sputtered on a 15 × 15 mm^2^ clean glass coverslip by using a magnetron sputtering system (Q300T D, Quorum, Laughton, East Sussex, UK). This system was endowed with a quartz microbalance for fine control of the sputtered film thickness and two sputtering heads, allowing sequential sputtering of two metals without breaking the vacuum. Therefore, a 30 nm Ag film was deposited on the Au film. At this stage, the Ag layer appears quite flat at a nanoscale (roughness ~2 nm). Thus, an inductively coupled plasma (ICP) treatment was applied to the Ag film to induce the formation of plasmonic nanostructures. In particular, the film was exposed to a synthetic air-based plasma for 90 s at a RF power of 18 W in an ICP chamber (PDC-32G-2, Harrick Plasma, Ithaca, NY, USA). As a side effect, this treatment leads to the oxidation of the Ag film (mainly oxidized to AgO) and the consequent loss of plasmonic activity. However, the pristine metallic character of the film was restored via a further 50 s plasma treatment in Ar atmosphere that led to the reduction of the AgO layer, preserving the former nano-patterning. This latter exhibits a coral-like nanomorphology with a strong porous character that provides a wide spectral response. Furthermore, the high porosity also entails a high density of sites where the amplification of the optical near field is particularly large (hot spots). In the following, such substrates will be indicated as c-SERS. Once prepared, c-SERS substrates were stored in a vacuum until use in order to avoid contamination. However, before use, SERS substrates were abundantly rinsed with distilled water to remove any possible contamination resulting from the handling of silver nano-structuration during the preparation protocol.

### 3.3. Functionalization of c-SERS Substrates for the Detection of LPS

The procedure followed for the detection of LPS is schematized in Figure 7. To assure minimal fluctuations in the SERS signal intensities due to the inter-batch reproducibility of the substrates, the experiment was conducted on the same substrate. In this regard, the c-SERS substrate was previously divided into 9 wells of the same area by thermal gluing a parafilm (Parafilm M, Bemis Company Inc., Neenah, WI, USA) mask (Step 1). After that, a 25 μL droplet of 10 μg/mL antibody anti-LPS diluted in Na-phosphate buffer, 50 mM, pH 7.4, was deposited in each well (Step 2). The incubation was performed for two hours at RT in a humid chamber, followed by washing (0.05% Tween-20 in water). Then, the substrate wells were incubated with a blocking solution (1% BSA, 1% sucrose, and 0.05% Tween-20 in 50 mM Tris-HCl buffer, pH 7.4, 35 μL per well) in order to avoid possible unspecific binding events of spurious molecules (Step 3). The incubation was performed for two hours overnight at +4 °C. In the following, c-SERS substrates coated by only Ab and the blocking solution (BS) will be indicated as Ab@c-SERS. After a washing step with Na-phosphate buffer (two times for 10 min under socking), eight wells were filled with LPS antigen diluted in milliQ water at different concentrations ranging from 10 ng/mL to 10 µg/mL (under critical micellar concentration [1]) and incubated at RT for one hour (Step 4). The remaining free well of LPS deposition was used as the control. All SERS measurements were performed within 24 h of the final incubation with LPS.

### 3.4. Raman System and Signal Acquisition

Raman analysis was performed using the WiTec Alpha 300 confocal microscope (WITec GmbH, Ulm, Germany), equipped with a Raman probe at 532 nm. This latter was focused on the sample by using a 20× dry objective (Plan N, NA 0.4, Olympus, Shinjuku, Tokyo, Japan), providing a 1.4 µm^2^ waist on the sample. Inelastically scattered photons were collected in a back-scattering geometry and, after filtering by an edge filter, sent via a 100 μm core fiber to a 600 g/mm grating of a high-throughput spectrometer. Spectra were acquired with a thermoelectrically cooled CCD camera (T = −60 °C) and pre-processed using tools available in the WiTec Project program (Version 2.08, WITec GmbH, Ulm, Germany). All SERS measurements were performed at room conditions. In particular, the relative humidity was around 20%, while the ambient temperature was around 25°. The acquired spectra were cleaned from the contribution of cosmic rays; therefore, a fourth-order polynomial background was subtracted.

### 3.5. Principal Component Analysis

The principal component analysis (PCA) is a well-consolidated multivariate statistical tool used to analyze multidimensional data sets (such as Raman spectra), highlighting the main difference among data points. It consists of an orthogonal linear transformation of the initial data set, which is projected into a new coordinate system describing the variability among data points. The first coordinate, called the first principal component (PC1), exhibits the maximum variance in the dataset; the following components consider the residual variance, and so on. The outcomes of the application of Raman/SERS spectra are represented by scores and loadings: scores represent the coordinates in the new space, while loading vectors describe the spectral features responsible for differentiation among samples. Herein, PCA was performed using a home-made Matlab program (Version R2021a, Natick, MA, USA). Before data deconvolution, performed by using Matlab routines for PCA, SERS spectra were pre-treated by using a custom-made routine developed to eliminate spurious cosmic ray contributions and to subtract a fourth-order polynomial background contribution.

## 4. Conclusions

In this work, we developed a label-free SERS-based LPS detection approach that combines the benefits of plasmonic enhancement with the selectivity provided by using a specific antibody. The method allows one to identify the SERS signal of the endotoxin and specifically quantify the amount of LPS on a single spot of the functionalized surface. The main advantages of this approach relative to the existing methods are a simple functionalization procedure, a broad detection range, and a low detection limit. In fact, the detection ability of the SERS substrate is validated by determining a LOD and LOQ of 12 ng/mL and 41 ng/mL, respectively. Importantly, the proposed technology could be easily expanded for the determination of other biological macromolecules.

## Figures and Tables

**Figure 1 ijms-24-12099-f001:**
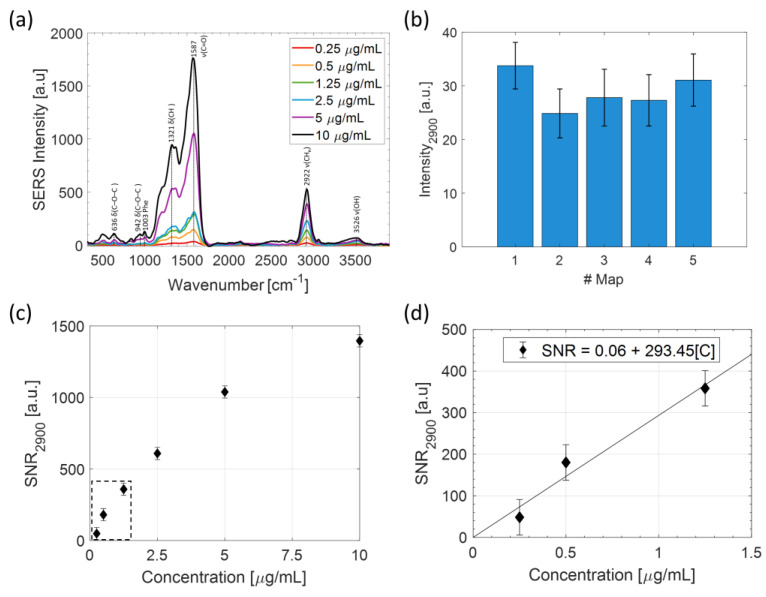
(**a**) Comparison of average SERS spectra acquired at different concentrations of LPS. Spectra were background corrected by a second-order polynomial curve. (**b**) Bar graph showing the average intensities and standard deviation of the peak at 2900 cm^−1^ obtained in 5 SERS maps of the well corresponding to [LPS] = 1.25 µg/mL. (**c**) Plot of SNR of the peak centered at 2900 cm^−1^ versus LPS concentration. Standard errors are included in the size of the experimental points. (**d**) Linear fit performed in the low concentration range used to estimate the limit of detection in correspondence with a SNR = 1.

**Figure 2 ijms-24-12099-f002:**
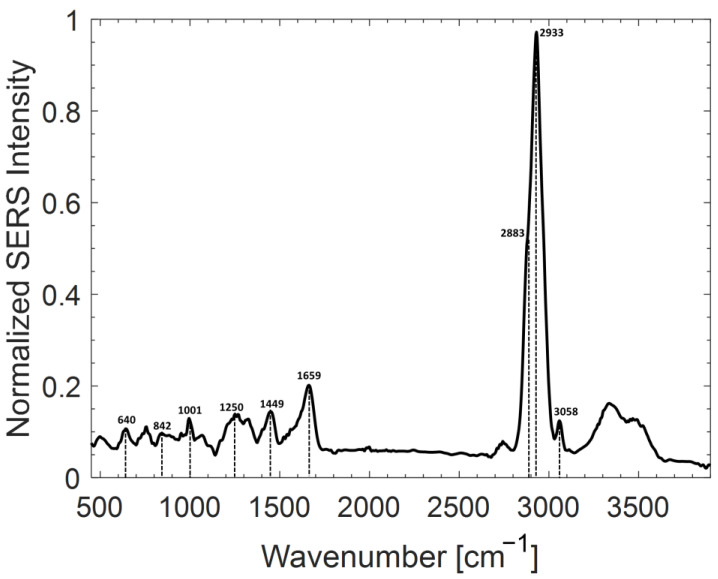
Average SERS spectrum of Ab (10 µg/mL) + BS deposited on the silver substrate, according to the procedure described in the text. Spectrum was obtained using laser power impinging on the substrate of 100 µW and an integration time of 1 s.

**Figure 3 ijms-24-12099-f003:**
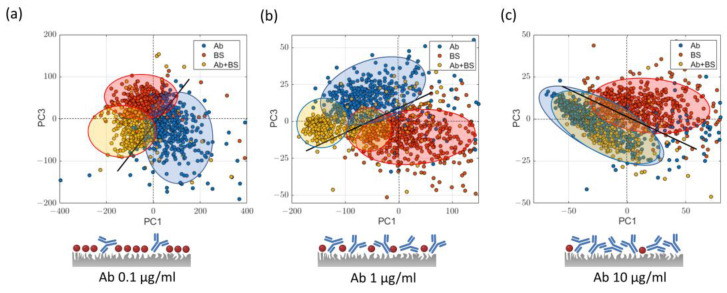
Principal component analysis of Ab@c-SERS substrate. Score plots obtained for the samples (**a**) “Ab + BS”, (**b**) “Ab”, and (**c**) “BS” as reported in the text. For “Ab+BS” sample, the Ab concentrations were 0.1, 1, and 10 µg/mL. In the lower panel, we report three cartoons, which illustrate the coating degree of Ab (BS molecules are represented by red circles).

**Figure 4 ijms-24-12099-f004:**
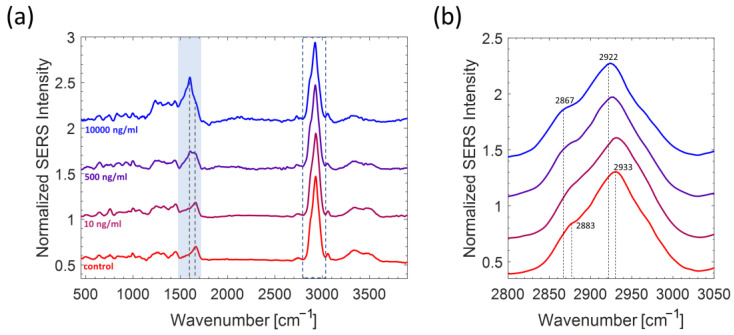
(**a**) Averaged SERS spectra of Ab@c-SERS upon exposure to LPS at different concentration values (for the sake of simplification, we have selected only some of the concentration values analyzed). Each spectrum, normalized to the prominent peak at ~2900 cm^−1^, corresponds to the average of 400 spectra acquired by raster scanning. The blue area highlights the most relevant spectral changes induced upon LPS binding to Ab@c-SERS that were observed. (**b**) Zoomed image of the dashed area reported in part (**a**) that highlights the spectral shifts of the band envelope maxima in the high frequency region.

**Figure 5 ijms-24-12099-f005:**
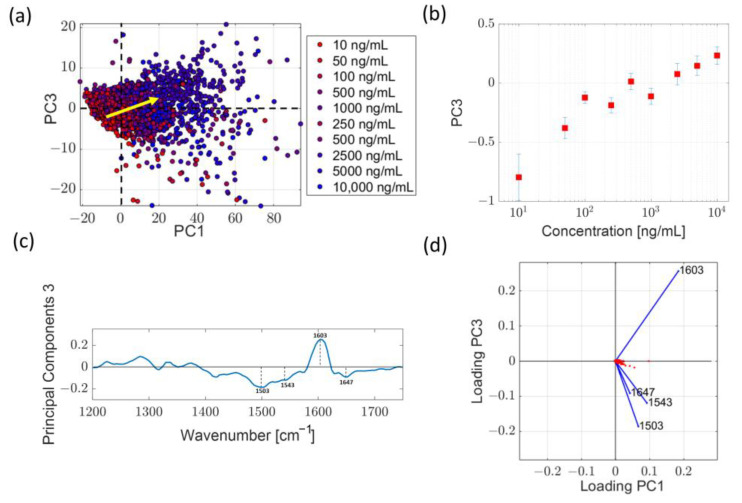
Principal component analysis of spectra at different LPS in the 1100–1800 cm^−1^ spectral region. (**a**) Projection of the score plot in the PC1-PC3 plane. Spectra due to samples at different LPS concentration values are represented by dots of different colors, as specified in the legend. The yellow arrow indicates the direction of the apparent shift of the score clouds observed as the LPS concentration values increase. (**b**) Trend of the projections along PC3 axes of the centroids of score cloud as a function of LPS concentration value. (**c**) PC3 loadings resulting from the PC analysis previously described. (**d**) Loading plot in the PC1-PC3 plane, highlighting the direction of variation of significant features in the PC3 vectors.

**Figure 6 ijms-24-12099-f006:**
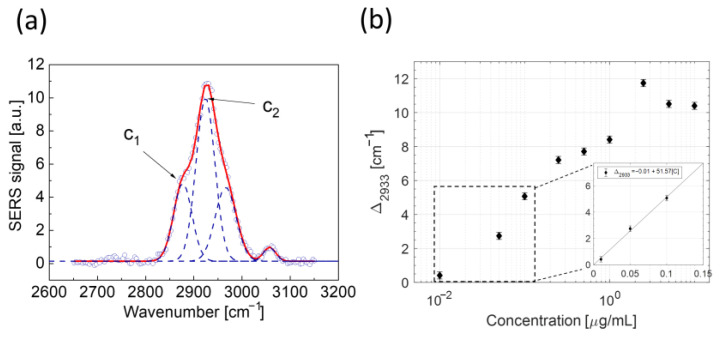
(**a**) Outcome of the fitting procedure of the spectral envelope in the region around 3000 cm^−1^ with four Gaussian peaks (blue dashed lines). The fitted envelope is reported as a continuous red line. (**b**) Trends of Δ_2933_ as a function of the LPS concentration value. The inset also reports the line resulting from a linear regression of data in the low concentration region in linear scale.

**Figure 7 ijms-24-12099-f007:**
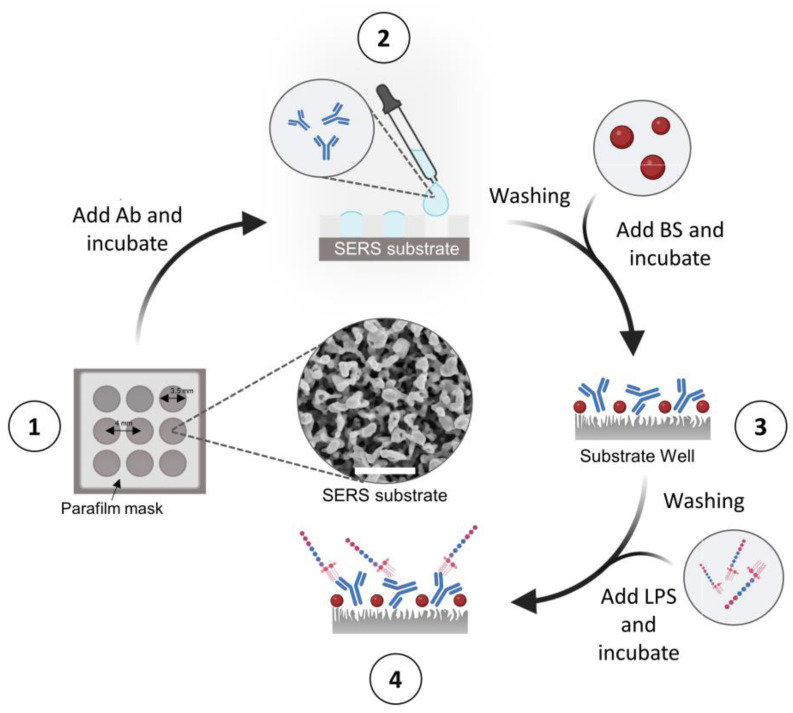
Schematic illustration of the procedure followed for the SERS detection of LPS at different concentrations. Step 1: Preparation of c-SERS substrate and wells setting up, using a parafilm mask. The inset shows a SEM image of coral-like SERS substrate (scale bar: 500 nm); Step 2: Immobilization of Ab by physical adsorption on each surface well; Step 3: Adding of BS to each surface well; Step 4: Adding of LPS to the functionalized SERS surface wells.

## Data Availability

Other data that support the findings of this study are available upon request from the corresponding author.

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
