# Peer review of "A Surface-Enhanced Raman Spectroscopy-Based Biosensor for the Detection of Biological Macromolecules: The Case of the Lipopolysaccharide Endotoxin Molecules"

_ijms, 2023, doi:10.3390/ijms241512099_

Round 1

Reviewer 1 Report

Lipopolysaccharides (LPS) are critical toxic substances that are responsible for food-related poisoning and health problems. In present study, the authors developed a label-free SERS-based approache for detection of LPS detection.

Some comments:

1.      Authors are requested to included detailed statistical information on the novel technology, such LOD, LOQ, and statistical errors (significance values).

2.      In Figure 1, authors could use some dark colors to depict the spectral lines. Light colors are not visible clearly from distant.

3.      Figure 3 could be enlarged with higher resolution.

4.      The units for liter is capital ‘L’ and should use according in micro or mill, etc.

5.      I recommend authors to consider some other relevant studies (for improving introduction and discussion) on application of SERS for biomolecule detection for example, RNA and DNA: Lee, S., Kadam, U. S., Craig, A. P., & Irudayaraj, J. (2013). In Vivo Biodetection Using Surface-Enhanced Raman Spectroscopy. Biosensors Based on Nanomaterials and Nanodevices, 157; PMID: 28602750; PMID: 28301688; PMID: 24460907; and PMID: 24631541.

6.      Figure 7 is good scheme; however, its resolution needs to be improved.

7.      Again, the conclusion needs to have key statistical information.

Language is fine.

Author Response

Response to Reviewer 1 Comments

Reviewer 1

Lipopolysaccharides (LPS) are critical toxic substances that are responsible for food-related poisoning and health problems. In present study, the authors developed a label-free SERS-based approache for detection of LPS detection. Some comments:

Point 1: Authors are requested to included detailed statistical information on the novel technology, such LOD, LOQ, and statistical errors (significance values).

Response 1: I would like to thank the reviewer for this suggestion. I have included information on the LOQ in the revised manuscript (please, see lines 309-310).

Point 2: In Figure 1, authors could use some dark colors to depict the spectral lines. Light colors are not visible clearly from distant.

Response 2: I would like to thank the reviewer for this comment. Figure 1 has been modified in the revised manuscript.

Point 3: Figure 3 could be enlarged with higher resolution.

Response 3: Following the reviewer ‘s suggestions the resolution of all figures in the paper have been improved.

Point 4: The units for liter is capital ‘L’ and should use according in micro or mill, etc.

Response 4: Following the reviewer ‘s suggestions I have checked the correctness of the reported units throughout the paper.

Point 5: I recommend authors to consider some other relevant studies (for improving introduction and discussion) on application of SERS for biomolecule detection for example, RNA and DNA: Lee, S., Kadam, U. S., Craig, A. P., & Irudayaraj, J. (2013). In Vivo Biodetection Using Surface-Enhanced Raman Spectroscopy. Biosensors Based on Nanomaterials and Nanodevices, 157; PMID: 28602750; PMID: 28301688; PMID: 24460907; and PMID: 24631541.

Response 5: Following the reviewer’s suggestion the manuscript bibliography was improved by adding the relevant suggested paper (please, see ref. [17]).

Point 6: Figure 7 is good scheme; however, its resolution needs to be improved.

Response 6: Following the reviewer ‘suggestion the Figure 7 resolution has been improved..

Point 7: Again, the conclusion needs to have key statistical information.

Response 7: I would like to thank the reviewer for this comment. In order to fix this the LOD and LOQ have been reported also in the conclusions paragraph.

Reviewer 2 Report

Dear authors,

It is a highly interesting paper, and the result should be of interest to many readers. However, one concern I have has to do with how results may depend on various environmental factors such as

(1) SERS fabrication conditions (ICP treatment duration, plasma treatment in Ar, rinsing by distilled water to remove contamination.)

(2) Pretreatment (age of the substrate, incubation with antibody and BS)

(3) SERS measurement (humidity of the environment)

I am sure these issues will be studied in the future, but for now I am very interested in knowing how you arrived at the measurement condition of 100 spectra recorded in a 20 um x 20 um with averaging over a total of 500. Would it be possible to show results from three separate 20 um x 20 um areas and demonstrate that results are the same?

Another minor issue I take is the claim that the reported method has much costs and time saving advantages over traditional methods. For example, I would imagine that you would have to prepare fresh Ab/BS SERS substrates each time before measuremnt, that would require at least two steps of incubation, each lasting two hours. Do you have data showing that SERS substrates so prepared can retain its performance at least for a few days?

In short, could you provide

(1) data from three separate 20 um x 20 um areas,

(2) data on any aging effect of Ab/BS SERS substrates, maybe up to a few days.

Just some minor issues

For example,

Line 31

"being the last the most conservative one among...."?

Author Response

Response to Reviewer 2 Comments

Reviewer 2

Point 1: It is a highly interesting paper, and the result should be of interest to many readers.

Response 1: I would like to thank the reviewer for this positive and encouraging feedback.

Point 2: However, one concern I have has to do with how results may depend on various environmental factors such as: SERS fabrication conditions (ICP treatment duration, plasma treatment in Ar, rinsing by distilled water to remove contamination.)

Response 2: I would like to thank the reviewer for this comment. SERS fabrication protocol have been deeply investigated in a previous paper [1]. For the sake of brevity, only the relevant parameters of the fabrication protocol have been reported in the manuscript (see 3.2 section in the “Materials and Methods” paragraph).

[1] Capaccio, A.; Sasso, A.; Rusciano, G. A simple and reliable approach for the fabrication of nanoporous silver patterns for surface-enhanced Raman spectroscopy applications. Sci. Rep. 2021, 11(1), 22295.

Point 3: Pretreatment (age of the substrate, incubation with antibody and BS).

Response 3: I would like to thank the reviewer for this question, which allowed us to improve the manuscript section regarding c-SERS substrate preparation. As now better specified in the text, once prepared, c-SERS substrates were stored in vacuum until use to avoid contaminations. However, before use, SERS substrates were abundantly rinsed by distilled water to remove any possible contamination resulting from the silver nano-structuration handling during the preparation protocol.

Point 4: SERS measurement (humidity of the environment)

Response 4: I would like to thank the reviewer for this question. All SERS measurements were performed at room conditions. In particular, the relative humidity was around 20%, while the ambient temperature was around 25°. This information has been added to the revised manuscript.

Point 5: I am sure these issues will be studied in the future, but for now I am very interested in knowing how you arrived at the measurement condition of 100 spectra recorded in a 20 um x 20 um with averaging over a total of 500. Would it be possible to show results from three separate 20 um x 20 um areas and demonstrate that results are the same?

Response 5: I would like to thank the reviewer for this question. We have added a further panel (panel b) in Figure 1. It reports the variability of the peak at 2900 cm-1 in 5 scans obtained in random regions of the substrate for the 1.25 µg/mL concentration.  As it is possible to see, the relative variability of the SERS signal on a single scan is around 19%. Notably, the average SERS intensities of each scan are consistent and by averaging them, the SERS signal fluctuation can be reduced to about 12%.

Point 6: Another minor issue I take is the claim that the reported method has much costs and time saving advantages over traditional methods. For example, I would imagine that you would have to prepare fresh Ab/BS SERS substrates each time before measuremnt, that would require at least two steps of incubation, each lasting two hours. Do you have data showing that SERS substrates so prepared can retain its performance at least for a few days? In short, could you provide. Data from three separate 20 um x 20 um areas,

Response 6: Following the reviewer’ suggestion the requested data have been shown in the revised Figure 1.

Point 7: Data on any aging effect of Ab/BS SERS substrates, maybe up to a few days.

Response 7: I would like to thank the reviewer, for surely raising an interesting and important issue, that deserves careful consideration. Now we have no answer to his/her question. In fact, all the measurements shown in the paper were performed within 24 hours from the final substrate incubation with the Ab+BS molecules. This latter information has been added in the revised manuscript. As argued by the reviewer, future investigations will address this issue.  

Point 8: Comments on the Quality of English Language. Just some minor issues. For example, Line 31 "being the last the most conservative one among...."?

Response 8: I would like to thank the reviewer for underlying this typo, which has been corrected in the revised manuscript.